# A new neuropeptide insect parathyroid hormone iPTH in the red flour beetle *Tribolium castaneum*

Jia Xie[1,2], Ming Sang[1], Xiaowen Song[1], Sisi Zhang[1], Donghun Kim[2,3], Jan A. Veenstra[4]*, Yoonseong Park[2]*, Bin Li[1]*

**1** Jiangsu Key Laboratory for Biodiversity and Biotechnology, College of Life Sciences, Nanjing Normal University, Nanjing, China, **2** Department of Entomology, Kansas State University, Manhattan, KS, United States of America, **3** Department of Applied Biology, Kyungpook National University, Sangju, Korea, **4** INCIA UMR 5287 CNRS, University of Bordeaux, Pessac, France

☯ These authors contributed equally to this work.
* jan-adrianus.veenstra@u-bordeaux.fr (JAV); ypark@ksu.edu (YP); libin@njnu.edu.cn (BL)

**Data Availability Statement:** Raw sequence Data of the RNA-seq analysis in this study are available in the Short Read Archive (SRA) database of NCBI with the accession numbers and NCBI URLs as

## Abstract

In the postgenomics era, comparative genomics have advanced the understanding of evolutionary processes of neuropeptidergic signaling systems. The evolutionary origin of many neuropeptidergic signaling systems can be traced date back to early metazoan evolution based on the conserved sequences. Insect parathyroid hormone receptor (iPTHR) was previously described as an ortholog of vertebrate PTHR that has a well-known function in controlling bone remodeling. However, there was no sequence homologous to PTH sequence in insect genomes, leaving the iPTHR as an orphan receptor. Here, we identified the authentic ligand insect PTH (iPTH) for the iPTHR. The taxonomic distribution of iPTHR, which is lacking in Diptera and Lepidoptera, provided a lead for identifying the authentic ligand. We found that a previously described orphan ligand known as PXXXamide (where X is any amino acid) described in the cuttlefish *Sepia officinalis* has a similar taxonomic distribution pattern as iPTHR. Tests of this peptide, iPTH, in functional reporter assays confirmed the interaction of the ligand-receptor pair. Study of a model beetle, *Tribolium castaneum*, was used to investigate the function of the iPTH signaling system by RNA interference followed by RNA sequencing and phenotyping. The results suggested that the iPTH system is likely involved in the regulation of cuticle formation that culminates with a phenotype of defects in wing exoskeleton maturation at the time of adult eclosion. Moreover, RNAi of iPTHRs also led to significant reductions in egg numbers and hatching rates after parental RNAi.

## Author summary

Vertebrate parathyroid hormone (PTH) and its receptors have been extensively studied with respect to their function in bone remodeling and calcium metabolism. Insect parathyroid hormone receptors (iPTHRs) have been previously described as counterparts of vertebrate PTHRs, however, they are still orphan receptors for which the authentic ligands

below: dsiPTHR1, (SRR8559987, https://www.ncbi.nlm.nih.gov/sra/?term=SRR8559987), dsiPTHR2 (SRR8559988, https://www.ncbi.nlm.nih.gov/sra/?term=SRR8559988) and control (SRR1176913, https://www.ncbi.nlm.nih.gov/sra/?term=SRR1176913).

**Funding:** This work was supported by the National Natural Science Foundation of China to BL (No. 31872970 & 31572326) (https://isisn.nsfc.gov.cn/egrantindex/funcindex/prjsearch-list?locale=zh_CN), the USDA National Institute of Food and Agriculture, Hatch project KS538 and National Institute of Health R21AI135457 to DK and YP (https://nifa.usda.gov/). The work is also supported by Nanjing Normal University Outstanding Doctoral Dissertation Cultivation Program to JX (YBPY18_001) (http://www.njnu.edu.cn/). The funders had no role in study design, data collection and analysis, decision to publish, or preparation of the manuscript.

**Competing interests:** The authors have declared that no competing interests exist.

and biological functions remain unknown. We describe an insect form of parathyroid hormone (iPTH) by analyzing its interactions with iPTHRs. Identification of this new insect peptidergic system proved that the PTH system is an ancestral signaling system dating back to the evolutionary time before the divergence of protostomes and deuterostomes. We also investigated the functions of the iPTH system in a model beetle *Tribolium castaneum* by using RNA interference. RNA interference of *iPTHR* resulted in defects in wing exoskeleton maturation and fecundity. Based on the differential gene expression patterns and the phenotype induced by RNAi, we propose that the iPTH system is likely involved in the regulation of exoskeletal cuticle formation and fecundity in insects.

## Introduction

Discoveries of neuropeptides and their receptors in various taxa in the postgenomic era have provided nearly comprehensive lists that provide crucial information for understanding their evolutionary processes and physiological functions. In addition, the development of new molecular techniques has rapidly expanded the knowledge of their functions in numerous cases of ancestral taxa. Homology-based searches in the existing genomic information are a powerful method; however, they may fail in cases where relatively rapid evolution occurs, thus leaving gaps in the knowledge due to punctuated equilibria in evolution. Functional studies of ancestral bilaterian neuropeptides have been successful by starting from the sequence similarities, e.g., vasopressin and thyrotropin-releasing hormone [1–4]

Vertebrate parathyroid hormone (PTH), the most important regulator of calcium ion homeostasis, and its receptors have been extensively studied in bone remodeling and calcium metabolism [5, 6]. Multiple PTH receptors in different phyla of vertebrates are known to be the consequence of multiple gene duplications and losses [7, 8]. Likewise, the ligand PTH also has undergone gene duplication and losses in vertebrates [9]. The PTH receptor lineage is traceable to before the time of the deuterostome-protostome split in the basal Bilateria [10–12], but there are no obvious homologous PTH ligands in the basal protostome lineages. Therefore, the PTH receptors in the basal lineages of Eukaryotes are still orphan receptors for which the authentic ligands and biological functions remain unknown. In insects, likewise, our previous study has described two receptors in the red flour beetle, *Tribolium castaneum*, as an orthologous group similar to the PTH-receptors of vertebrates in the G protein-coupled receptor (GPCR) B group [13, 14]. The ligand for these receptors has not yet been identified [15].

In this study, we have uncovered the authentic ligand for the insect PTH receptor (iPTHR). Our strategy for deorphanization of this ancestral GPCR utilized a genomic survey of numerous species in insects and further basal lineages. We looked for the similar patterns of taxonomic distributions of iPTHR and of neuropeptide-like sequences. We found that the taxonomic distribution of the neuropeptide previously described as PXXXamide (where X is any amino acid) in the cuttlefish *Sepia officinalis* (Mollusca) [16], is found in a mirror image of the distribution pattern of iPTHR. Specifically, the loss of iPTHR in Diptera and Lepidoptera coincided with the absence of this specific neuropeptide in the same taxa. We demonstrated that this previously uncharacterized neuropeptide is an active ligand on these receptors and we named the peptide insect PTH (iPTH). The phenotypes from RNA interferences (RNAi) and the RNAseq data suggest that the iPTH system is involved in the maturation of the exoskeletal cuticle in the wings at the time of adult eclosion.

## Results

### Evolution and diversity of iPTH and iPTHR in insects

In our survey of iPTHs, the neuropeptide originally described as PXXXamide [16], we identified iPTHs in many species of insect and in other arthropod taxa, but lacking in Diptera and Lepidoptera genome sequences. In *T. castaneum*, the gene encoding iPTH was found in the linkage group LG6 (Fig 1A) [17, 18]. The conceptual translation contains a signal peptide at the N-terminus, followed by a dibasic cleavage site KR (Lys-Arg) and the conserved mature peptide ending with putative amidation and a dibasic cleavage signal (GRR) with further extension of the C-terminal sequence. Although this mature peptide has no sequence homology to vertebrate PTH (lower than 12% identities with PTH, PTHrP, and Tip39 of *Homo sapiens*), highly conserved iPTHs were found in various insects and other taxa (Fig 1B and S1 Table). In the predicted mature peptides, N-terminal amino acids SD (Ser and Asp at positions 7 and 8 in the alignment Fig 1B) are highly conserved. The C-terminal P (Pro at the −4 position) for the original description of the peptide as PXXXamide was found to be mildly conserved in an expanded survey of the sequences. The C-terminal amidation and cleavage signal GXX (X = positively charged amino acids)[19] are strictly conserved in different species.

**A** LG6:2679258..2679999, LOC103313383 (TcasGA2_TC015123)

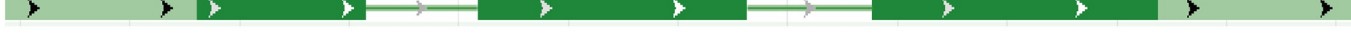

>iPTH *Tribolium castaneum*
MKTITFCFFVVLVMSVQNVFAGPRYRL**KR**VSDAHLADLQSRIALNNKLKGVSVTMPVGGGRIDPGRI**GRR** **RR**SQTRFLDVLFNQSEEDKGDPVELTDYENLIQRLLNLE

**B**

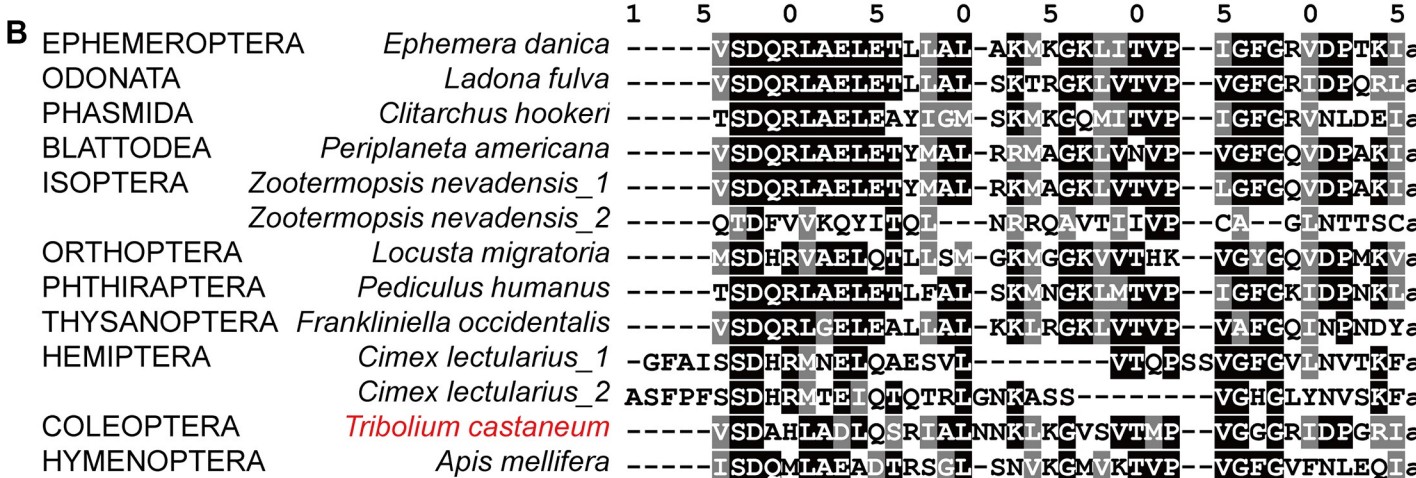

**C**

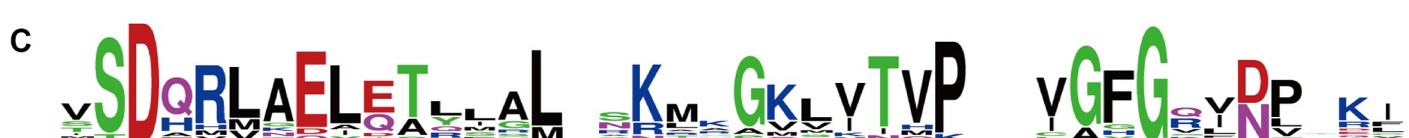

**Fig 1. Conserved iPTH and iPTHR sequences.** (A) The *iPTH* gene structure and the conceptual translation in *T. castaneum*. (B) An alignment showing the consensus of iPTH in various insect species. The letters with black backgrounds are identical and gray are similar amino acids with the 50% majority rule. (C) The sequence logo of the high to moderate levels of conserved residues in the predicted mature peptides in various insect species.

The iPTHRs belong to the class B1 G protein-coupled receptors (GPCRs) and are considered counterparts of vertebrate PTHRs based on sequence similarity (Fig 2A; S1 Fig). The evolutionary processes of vertebrate PTHRs have previously been described as the events of multiple gene duplications and deletions [7, 8]. We also found support for gene duplication events of iPTHR in the insect lineages (Fig 2A; S1 Fig). The genome sequences of many insect species revealed two or more copies of the PTH receptor, although it is difficult to resolve whether the gene duplication was a single event or independent multiple events because of incongruent branching patterns of the tree constructed based on the amino acid sequences. In *T. castaneum*, the genome data revealed two iPTHRs at LG4 and LG3 for *Tc-PTHR1* and *R2*, respectively. There were no iPTHR homologs in Diptera and Lepidoptera genome sequences, consistent with the absence of iPTH sequence in those taxa (S1 Fig; S1 Table).

## Functional expression of PTH receptors and iPTH activity assays

The *Tribolium* gene annotations (Tcas5.2) [18, 20] were used to obtain the open reading frames of the *Tc-iPTHR1* (TC008110) and *R2* (TC010267) by polymerase chain reaction (PCR) (GenBank with the accession numbers XM_008193931.2 and XM_964860.3, respectively) (S2A Fig). Conservation of seven cysteines (marked as stars on top of the alignment in S2B Fig) in the predicted extracellular loops including the N-terminal extracellular region implies a significant role of disulfide bridges for formation of the ligand binding pocket, as stabilization of extracellular structure by disulfide bonds in GPCR-B has previously shown [21–24]. The C-terminal region showed large diversity in the sequences compared to the transmembrane regions. The iPTHRs were functionally expressed in a Chinese hamster ovarian (CHO) cell line to which components of aequorin as the responder to calcium mobilization in CHO cells had been added, which is an approach that has been described before [25, 26]. We found that the CHO cells transfected for expression of either receptor, Tc-iPTHR1 or Tc-iPTHR2, were activated by chemically synthesized Tc-iPTH. Tc-iPTHR1 is ~7× more sensitive ($EC_{50}$ = 63 nM) than Tc-iPTHR2 ($EC_{50}$ = 413 nM). The cells without the Tc-iPTHR constructs did not respond to the Tc-iPTH at 10 μM (Fig 2B).

## Expression patterns of *Tc-iPTH* and *Tc-iPTHRs*

Transcript levels of the genes, *Tc-iPTH*, *Tc-iPTHR1*, and *Tc-iPTHR2*, were investigated to understand spatial and temporal expression patterns. Quantitative real time reverse transcriptase PCR (Q-PCR) revealed high levels of the transcripts in late larval and late adult stages for Tc-*iPTHR1*, the late adult stage for *Tc-iPTHR2*, and the late pupal stage for *Tc-iPTH* (Fig 3). The gut and central nervous system (CNS) had high transcript levels of all three genes in both the larva and adult stages (Fig 3D to 3I). Further investigations with separate foregut, midgut, and hindgut in larval, pupal, and adult stages found that the transcript levels were highest in the pupal foregut and midgut (S3A and S3B Fig). Interestingly, the *Tc-iPTHR2* transcript was found to be moderately expressed in adult elytra and epidermis (Fig 3H). The *Tc-iPTHR2* transcript was further confirmed by Q-PCR for the samples of pupal and adult elytra, and we found that adult elytra, but not pupal elytra, showed a moderate level of the *Tc-iPTHR2* transcript (S3D Fig).

Immunohistochemistry of Tc-iPTH revealed a number of neural cells and axonal projections in the central nervous systems of larval (Fig 4A to 4G, 4R and 4S), and pupal ganglia (Fig 4H to 4J, 4R and 4S). Visualization of cell bodies in the Tc-iPTH staining rarely occurred while the neural projections always appeared. Inconsistent immunohistochemical staining patterns of cell bodies depending on the samples were common for various neuropeptides in *T. castaneum* including Natalisin [25]. Combining the results from multiple samples (n>50 in each

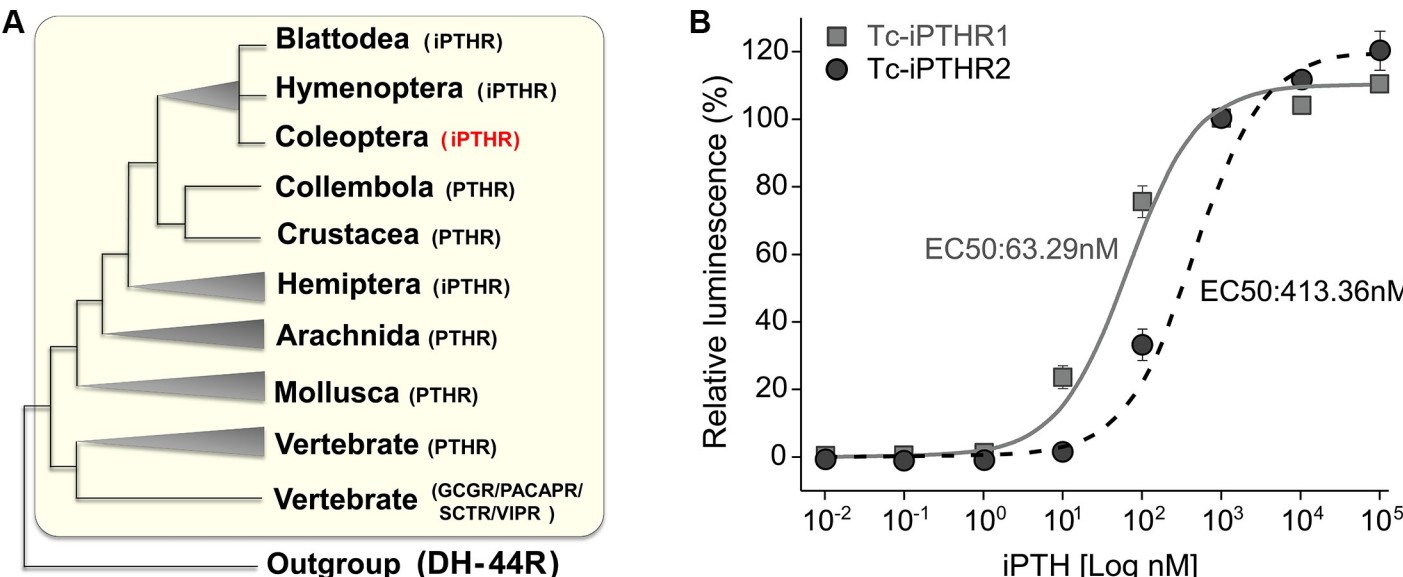

**Fig 2.** (A) Phylogenetic tree of the PTHRs of the metazoan species constructed by the Maximum likelihood method with the JTT model (see more details in S1 Fig for 1000 bootstrapping tests using MEGA 7.0 software). (B) Dose-response curves of the iPTH on the iPTHRs expressed in the CHO cells. The EC50 of Tc-iPTH to Tc-iPTHR1 and Tc-iPTHR2 were 63.29 nM and 413.36 nM, respectively.

larvae and pharate adults), we concluded that there are at least two segmental ganglionic cells for Tc-iPTH expression, including the brain, subesophageal ganglion, and other segmental ganglia (Fig 4G to 4J and 4S). Two pairs of dorso longitudinal projections connected these varicosities (Fig 4D and 4E), while in some cases, transverse projections of neurons from ganglia were also observed (Fig 4B, 4C and 4S). The brain had extensive varicosities in the protocerebral and deutocerebral regions in both larva (Fig 4G and 4R) and pupa (Fig 4H and 4R).

Strong immunoreactivities were also found in the gut (Fig 4K to 4Q, 4T and 4U) including stomatogastric nervous projections and the puncta surrounding the projections on the anterior half of the midgut ran along the longitudinal axis (Fig 4K and 4T), which are similar to the bipolar enteric plexus cells in *Manduca sexta* [27]. Immunoreactive stomatogastric nerves and the puncta were also observed in the pharate adults (Fig 4N and 4Q). A separate neural projection was found in the anterior part of the ileum (Fig 4L, 4M and 4Q). In addition to the neural projection in innervation of the guts, enteroendocrine cells in the gut were also immunoreactive in both larva and pharate adults (Fig 4K, 4L, 4M, 4P and 4T). Large numbers of these cells were shown in the anterior and posterior part of the midgut (Fig 4T and 4U), while a small number of immunoreactive enteroendocrine cells were also observed in the middle part of the gut. Pharate adults also had positive reactions in very small enteroendocrine cells (<~3 μM in diameter) (Fig 4P). Preimmune serum used as the negative control did not show any specific staining patterns (S4A to S4G Fig).

Immunohistochemistry of the receptor Tc-iPTHR1, which has shown a strong activity by Tc-iPTH in the receptor-ligand activity assay, was carried out. Strong immunostaining was observed in the neural cells and axonal projections in the CNS in larval (Fig 5A to 5C), pupal (Fig 5D to 5F) and adult stage (Fig 5G to 5I). One or two pairs of the neural cells were consistently captured in the brain (Fig 5A and 5G) and thoracic ganglia (Fig 5B, 5D and 5H) in different individual samples. Immunoreactive projections also appeared in the thoracic ganglia (Fig 5B, 5E and 5H), as well as abdominal ganglia (Fig 5C, 5F and 5I). The immunoreactivity in the gut showed large stomatogastric neural cell bodies located on the anterior tip of midgut

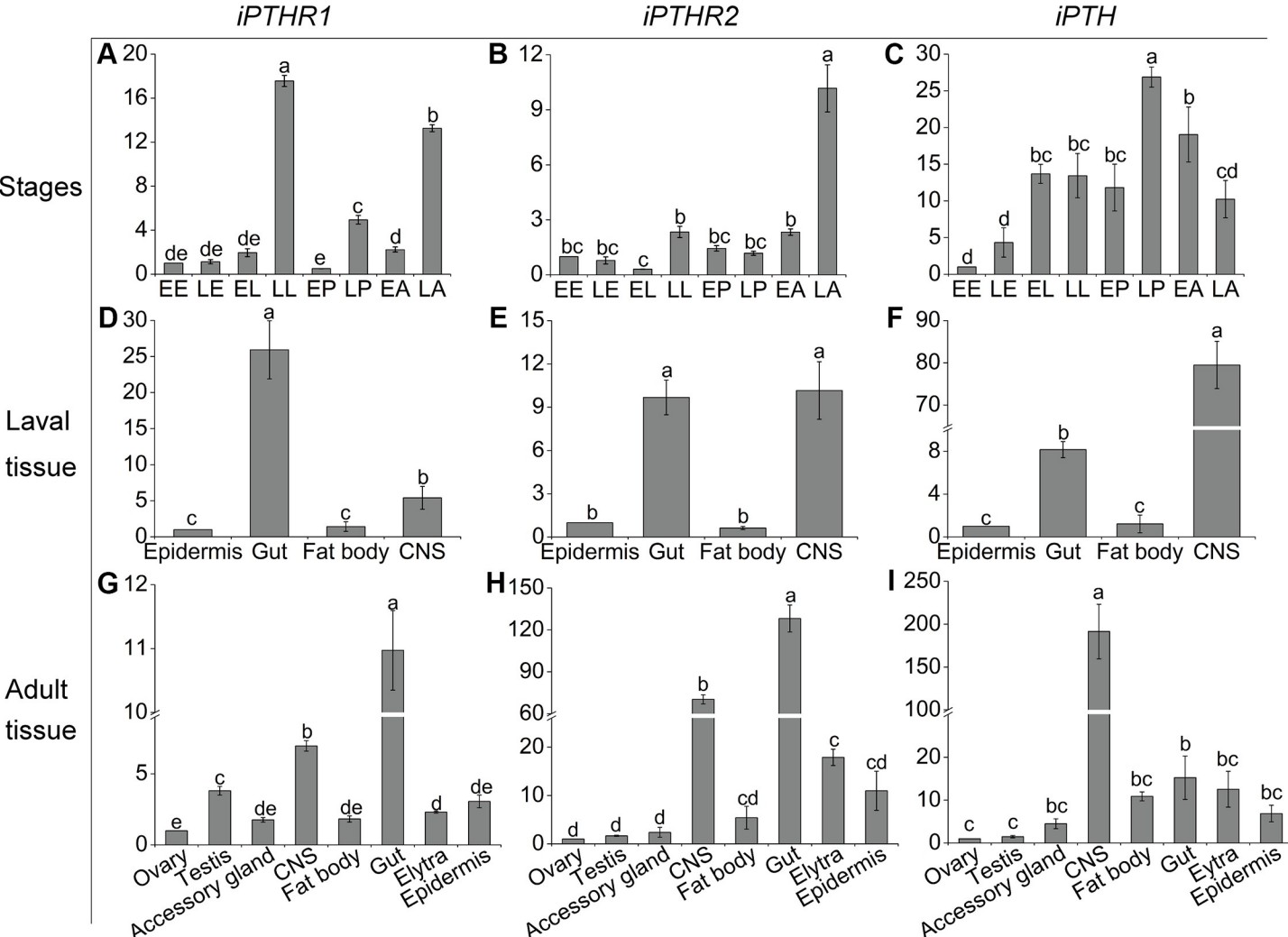

**Fig 3. Temporal and spatial expression patterns of *Tc-iPTH*, *Tc-iPTHR1*, and *Tc-iPTHR2* measured by Q-PCR.** (A-C) Expression patterns at different developmental stages. Different developmental stages are: EE, early embryos; LE, late embryos; EL, early larvae; LL, late larvae; EP, early pupae; LP, late pupae; EA, early adults; LA, late adults.(D-F) Expression patterns in various tissues from late larvae. The tissues are epidermis, gut, fat body and CNS (central nervous system). (G-I) Expression patterns in various tissues from late adults. The *Tribolium* ribosomal protein S3 (rps3) transcript was the internal control. The averages and standard errors are for three biological replicates. Different lowercase letters indicate statistically significant differences (p≤0.05).

(Fig 5K) and their projections running longitudinally on the midgut in the anterior half (Fig 5J, 5L and 5M). The staining pattern is similar to the immunoreactivity of the ligand Tc-iPTH, but generally thinner than the staining for Tc-iPTH. There were no immunoreactive enteroendocrine cells. Negative controls of the immunohistochemistry of anti-Tc-iPTHR1 antibody staining were carried out using the preimmune serum, which did not show any staining in the CNS and gut (S5 Fig). Since both antibodies, anti-Tc-iPTH and anti-Tc-iPTHR1, were raised from rabbit, we did not perform the double staining.

## RNAi of *Tc-iPTHRs* resulting in eclosion and wing defects

Both larval RNAi and pupal RNAi of *Tc-iPTHR1* and *Tc-iPTHR2*, which successfully suppressed the expression of their specific target genes (Fig 6A and 6D), resulted in partial defects

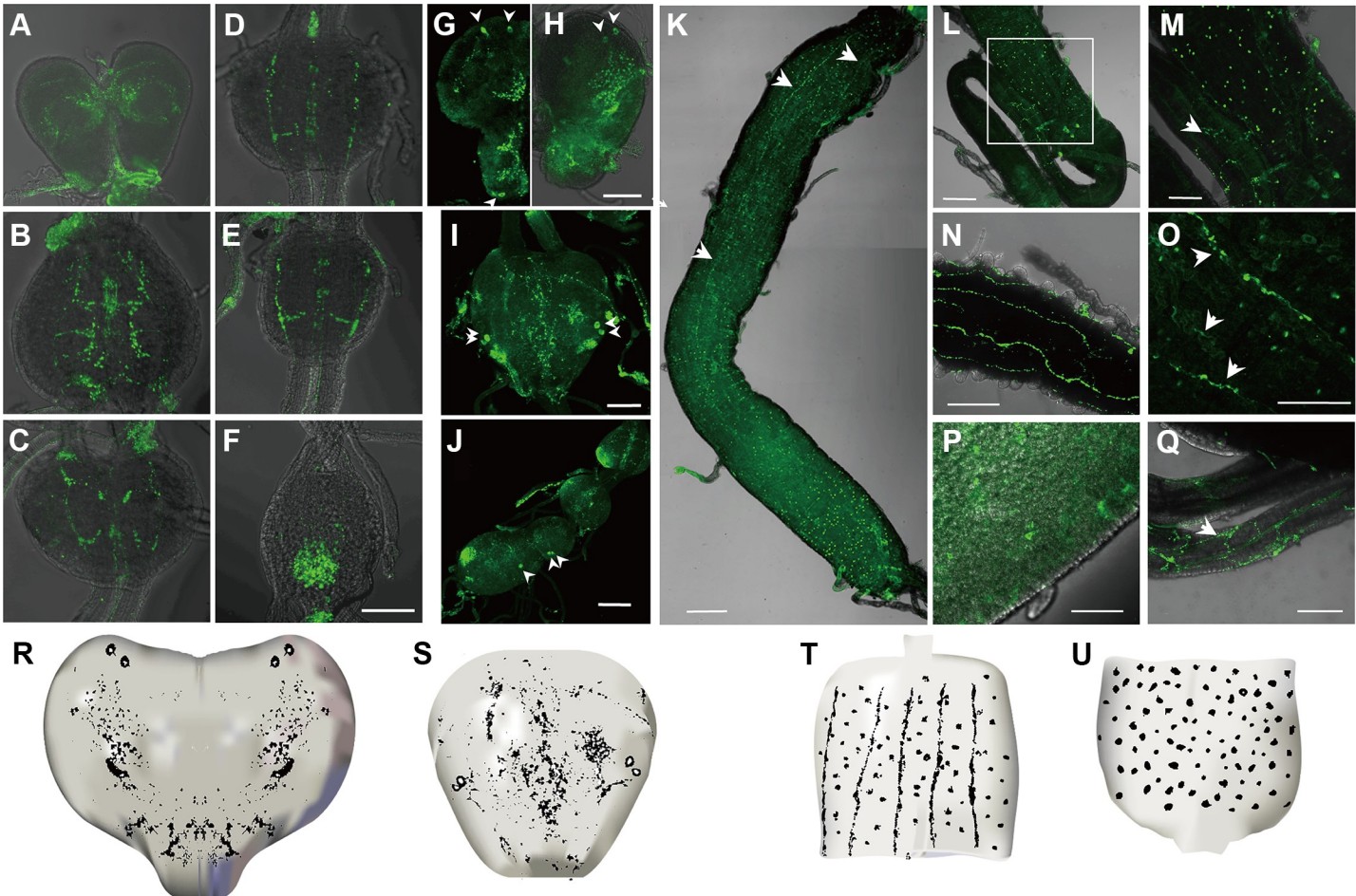

**Fig 4.** Immunohistochemistry of Tc-iPTH in the central nervous system (A-J) and gut (K-Q) in larval (A-G, K-M), and pharate adult (H-J, N-Q) stages. The arrowheads (G-J) indicate positive neural cells for anti- iPTH, and the white arrows (K, M, O, Q) indicate positive neuronal projections for anti- iPTH. Bars are 200 μm. Diagrams of the brain (R), ganglion (S) showed the neuronal cells and axonal projections in the CNS. Diagrams of anterior part of midgut (T) showed stomatogastric nervous projections. The posterior part of midgut (U) showed the enteroendocrine cells.

in adult eclosion (Fig 6B and 6C). The eclosion deficiencies were associated with incomplete shedding of the pupal cuticle, failure in spreading the wings, and irregularly wrinkled elytra (Fig 6G; Fig 7). RNAi of *Tc-iPTHR1* in the larval stage resulted in 81.3±8.4% of individuals with ecdysial arrest at the time of adult eclosion (Fig 6B; Fig 6G) and 14.3±5.8% showing adult morphological deficiency (Fig 6B; Fig 7A). *Tc-iPTHR2* RNAi showed 43.8±10.5% individuals arrested at the time of eclosion (Fig 6B; Fig 6G), and 8.0±2.8% had adult morphological deficiency (Fig 6B; Fig 7A and 7B). RNAi with the dsRNA injections at the pupal stage resulted in the same phenotypes but with lower numbers of individuals affected (Fig 6C). The individuals with *Tc-iPTHR1* RNAi laid significantly fewer eggs, 2 eggs compared to 7 eggs per day in the control (Fig 6E). The percentages of hatching eggs were also significantly reduced to 10 and 40% in RNAi of *Tc-iPTHR1* and *Tc-iPTHR2*, respectively, compared to ~80% hatching rate in the control (Fig 6F).

However, the RNAi of the ligand *Tc-iPTH* did not show any phenotype (Fig 6B, 6C, 6E and 6F). Immunohistochemistry indicated that the suppression of *Tc-iPTH* by the RNAi may not have been sufficient to induce a phenotype (S4H to S4K Fig).

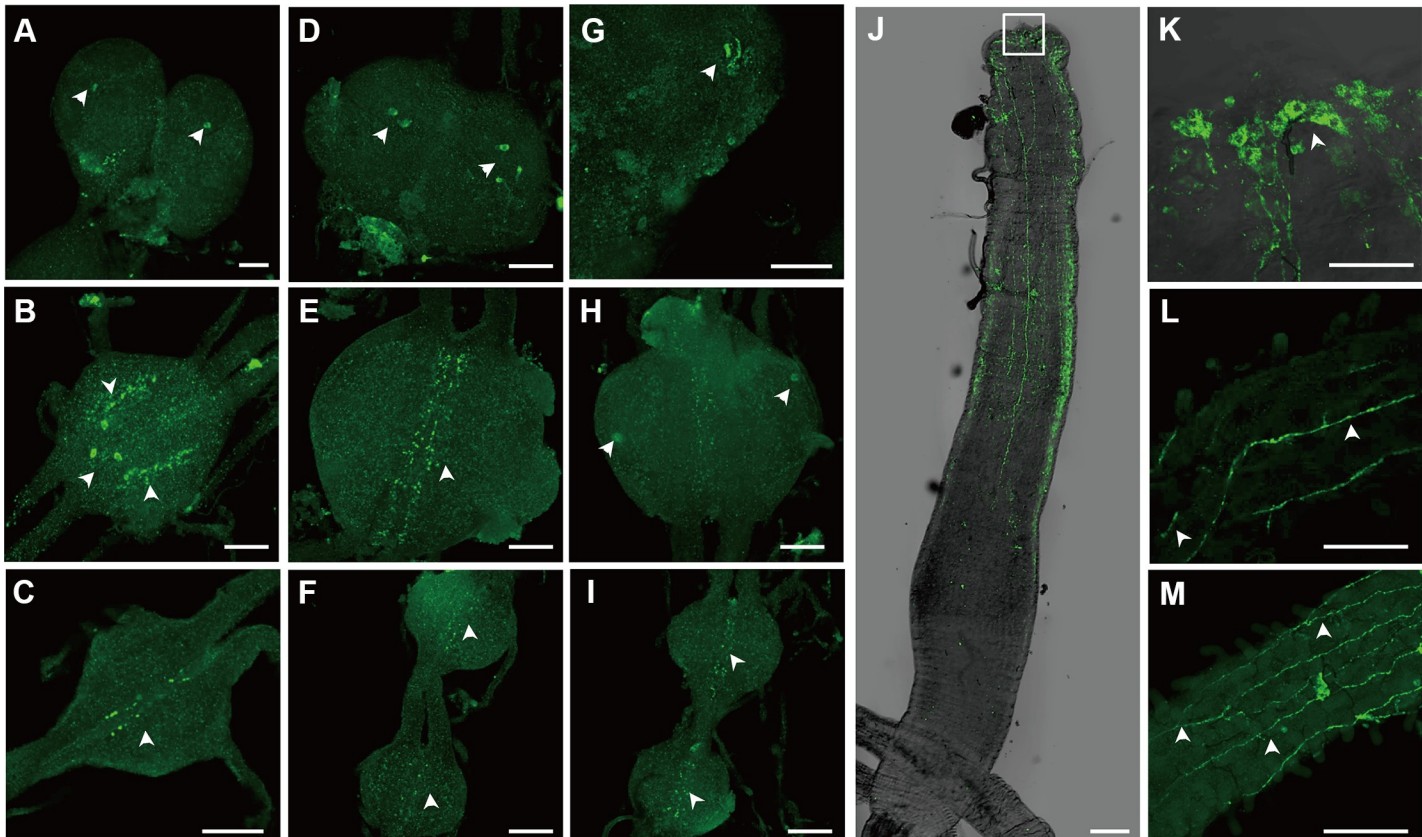

**Fig 5.** Immunohistochemistry of Tc-iPTHR1 in the central nervous system (A-I) and gut (J-M) in larval (A-C, J, K), pupal (D-F, L) and adult (G-I, M) stages. The arrowheads indicate positive neural cells or neuronal projections for anti-iPTHR1. Bars (A-I) are 50 μm and bars (JLM) are 200 μm.

## Gene expression changes induced by the RNAi of *Tc-iPTHR1* and *Tc-iPTHR2*

RNAseq of the individuals with the RNAi of *Tc-iPTHR1* and *Tc-iPTHR2* revealed large numbers of differentially expressed (DE) genes. Principal component analysis (PCA) of gene expression patterns in the control, ds*iPTHR1*, and ds*iPTHR2* showed that the RNAi of *Tc-iPTHR1* and *Tc-iPTHR2* resulted in highly correlated patterns of gene expression changes compared to the control (Fig 8A; S6 Fig). Likewise, three-dimensional scatter plots also showed that DE genes in the ds*iPTHR1* and ds*iPTHR2* treatments have similar patterns (Fig 8B). When the DE genes (>4× up or down) were separately plotted for the log$_2$ of the expression ratios, the correlation between ds*iPTHR1* and ds*iPTHR2* treatments was obvious ($R^2$ = 0.78) (Fig 8C). Large portions of DE genes were common in response to the ds*iPTHR1* and ds*iPTHR2* treatments (110 and 242 genes for >4× up or down genes, respectively, Fig 8D). Manual examination followed by computational annotation and gene ontology term enrichment tests of the list found some noticeable enriched gene groups in the list of DE genes.

To verify the RNAseq-based DE data, we randomly selected 22 genes for each treatment, and Q-PCR was performed. The Q-PCR results were highly correlated with the RNAseq-based analyses by having $R^2$ >0.96 in the linear regression (S7 Fig).

In the gene ontology (GO) term enrichment analyses for the DE genes, the largest portion encodes extracellular structural components including cuticular proteins; 68 are down- and 40 genes are up-regulated (Fig 8D; S1 Dataset and S2 Dataset). The 68 down-regulated genes

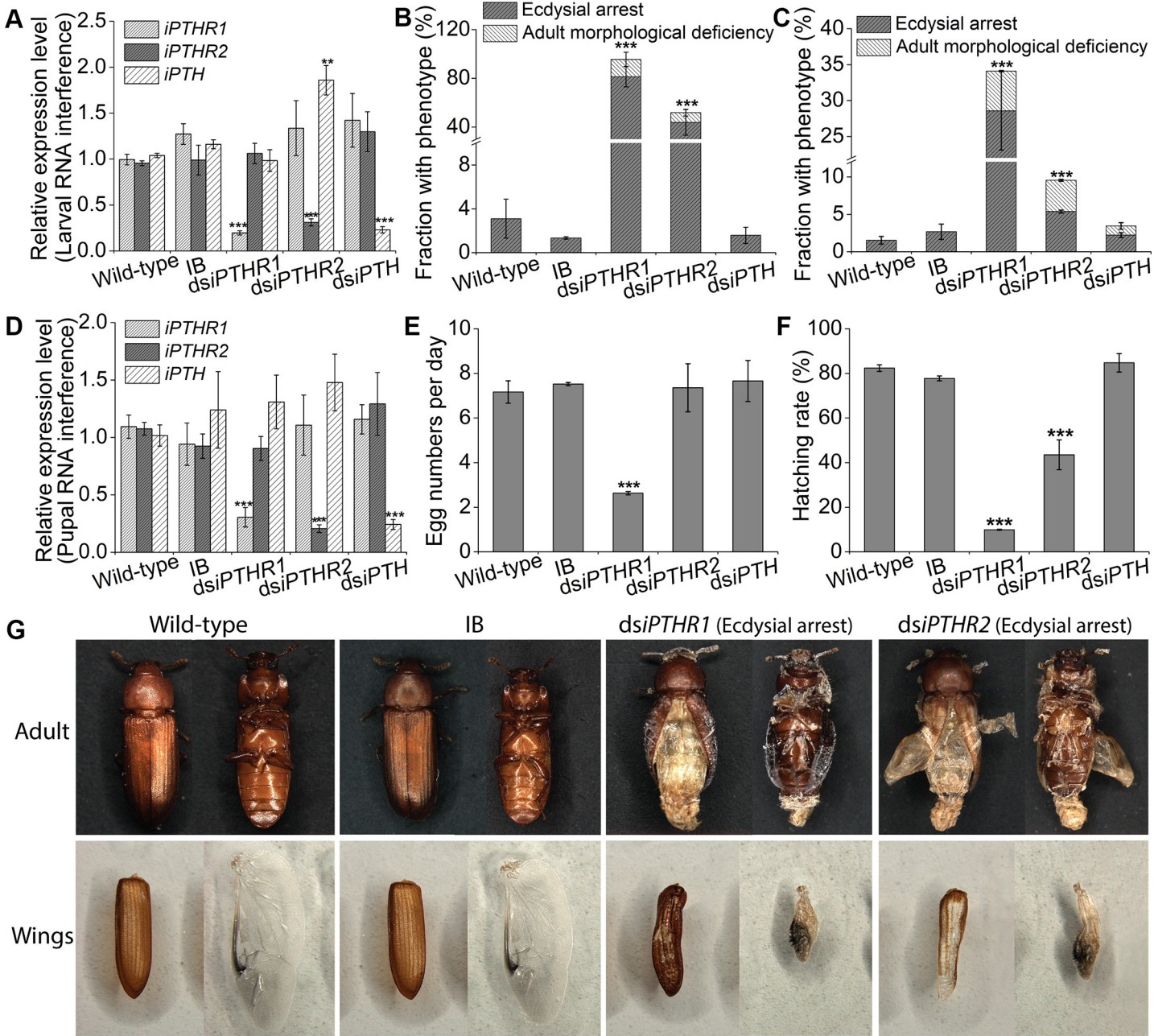

**Fig 6. RNAi results in suppression of the transcript levels with wing and fecundity phenotypes.** Different treatment group are wild-type, IB (buffer injection), ds*iPTH*, ds*iPTHR1* and ds*iPTHR2*. (A) Larval RNAi suppresses only the target transcripts. (B) The percentage with abnormal phenotypes in the beetles among all the animals after larval RNAi (C) and after pupal RNAi. (D) Pupal RNAi suppresses only the target transcripts. (E) Reduced egg numbers in RNAi of iPTHR1 and (F) hatching rate of the offspring after pupal RNAi. (G) RNAi phenotypes of the beetles arrested at the time of eclosion and abnormal wings. Both elytra and hindwings are shown.

were mainly cuticular proteins and included keratins and mucins, while the 40 up-regulated genes included 17 cuticle proteins and six different keratins and seven prismalin-14s. In the phylogeny of down-regulated cuticle proteins, it showed that most of the cuticle proteins belong to the CPR (Cuticular Protein with the Rebers and Riddiford Consensus) family, in which 10 of them belong to RR-1 cluster and 18 are in RR-2 cluster (S8 Fig).

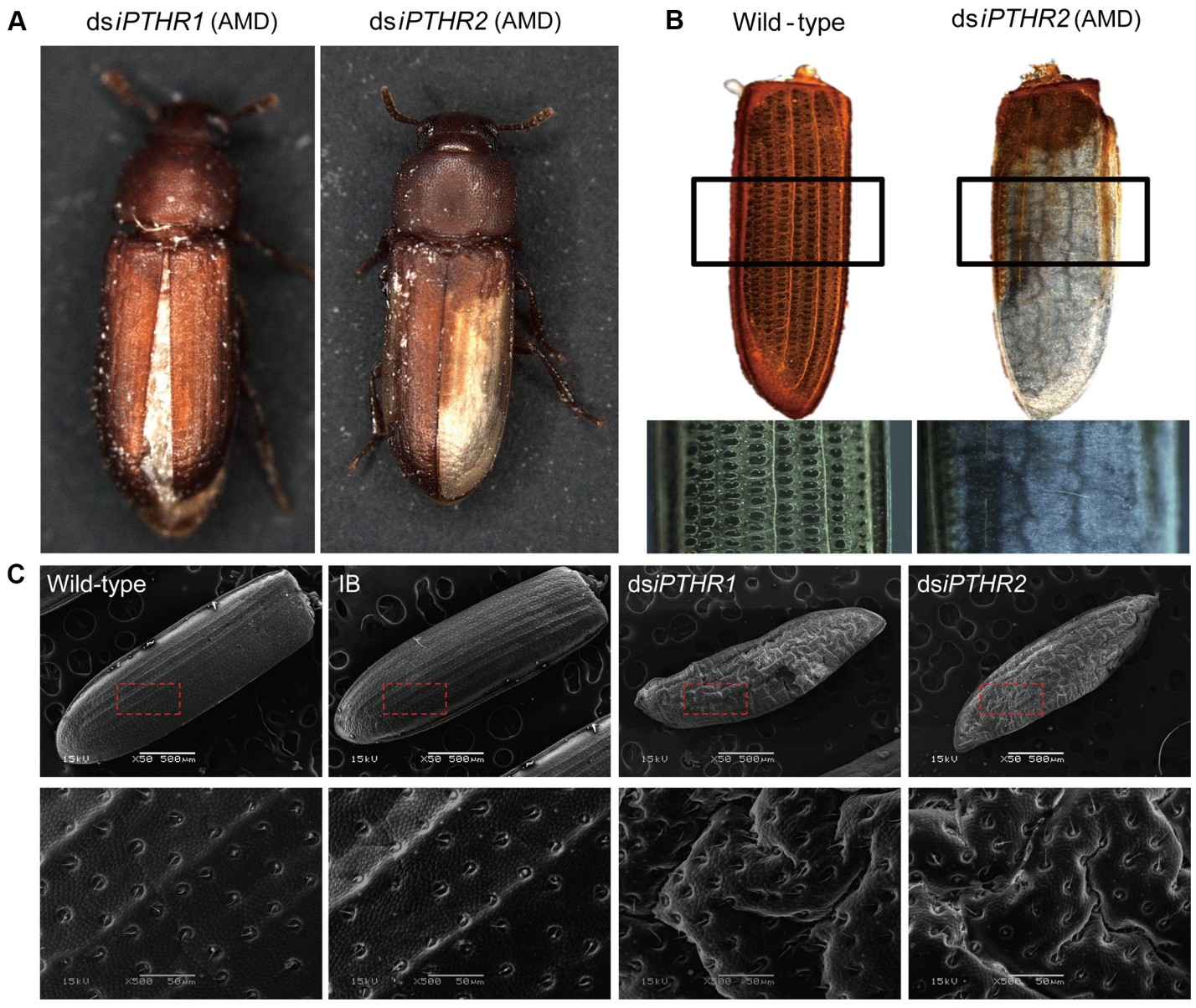

**Fig 7. Phenotypes after dsRNA treatments for *Tc-iPTHRs*.** The pictures are taken of the individuals who survived the dsRNA treatments but with some morphological abnormalities (adult morphological deficiency, AMD). (A) Survival after RNAi of *iPTHR1* showed incomplete closure of the elytra and rough elytra surface and survival after RNAi of *iPTHR2* often showed incomplete tanning of the elytra. (B) The elytra of wild-type adult and ds*iPTHR2* survival. (C) Scanning electron microscopy (SEM) of elytra after RNAi with the 10-day-old adults.

In addition, genes for cuticular metabolism were also captured, in which *chitinase 8* was up-regulated and six genes down-regulated (including *chorion peroxidase-like*, *chitinase 5*, *chitinase 10*, and *knickkopf3*). Three genes encoding cytochrome P-450 (*Cyp450*) were up-regulated, while five *Cyp450* genes including Halloween gene *spook* (*Cyp450 307A1*) were significantly down-regulated. Four different antimicrobial peptides were up-regulated. Down-regulated genes were ecdysone-induced *74EF*, *E78*, *DHR3*, *HR4*, *E3*, *neuropeptide F receptor*, and *rickets*, which codes the *bursicon receptor*. Six genes encoding sensory molecular components, gustatory receptor, odorant receptor, and chemosensory proteins, were also down-regulated.

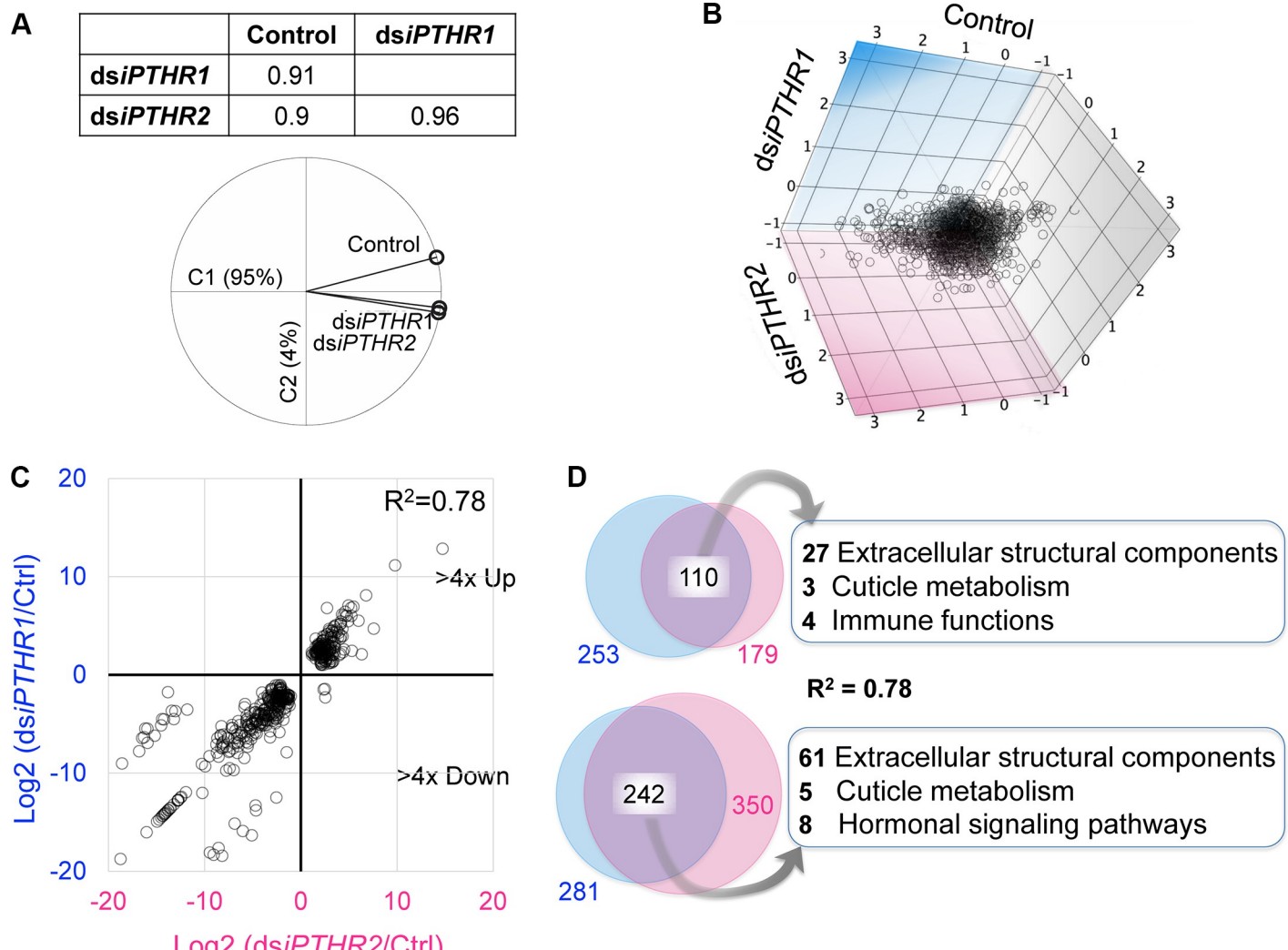

**Fig 8. RNAseq showing the similar pattern of changes in the expression of genes in ds*iPTHR1* and ds*iPTHR2* compared to the control.** (A) Principal component analysis of gene expression patterns in control, ds*iPTHR1*, and ds*iPTHR2*. (B) Three-dimensional scatter plots analysis of differentially expressed genes in contrast with the control, ds*iPTHR1* and ds*iPTHR2*. (C) Differentially expressed genes (>4× up or down) separately plotted for the log2 of the expression ratios. The expression changes in ds*iPTHR1* and ds*iPTHR2* share a similar pattern. (D) Venn diagrams show the common genes differentially expressed in ds*iPTHR1* and ds*iPTHR2* treatments.

## RNAi of *Tc-iPTHRs* increased chitin content but did not affect 20E amount

Based on the wrinkled elytra phenotype and the large numbers of genes encoding cuticle protein in the DEs after the RNAi of *Tc-iPTHR1* and *Tc-iPTHR2*, we measured the chitin content and 20-hydroxyecdysone (20E) in the control and in the ds*iPTHR1* and ds*iPTHR2* beetles. Significantly higher amounts of chitin, measured by the amount of hydrolyzed N-acetylglucosamine (GlcNAc), were found in the RNAi of *Tc-iPTHR1* (Fig 9A and 9B). More specially, chitin contents in the isolated elytra showed RNAi of both *Tc-iPTHR1* and *Tc-iPTHR2* were significantly higher than that in the control (Fig 9C).

The 20E titer was measured in two different developmental stages, 3-day-old and 5-day-old pupa. No significant effect on the 20E titer after RNAi of *Tc-iPTH*, *Tc-iPTHR1* or *Tc-iPTHR2* was found (S9 Fig), using a20E antibody that has a relatively high specificity against 20E (10× higher affinity for 20E than that for ecdysone) [28].

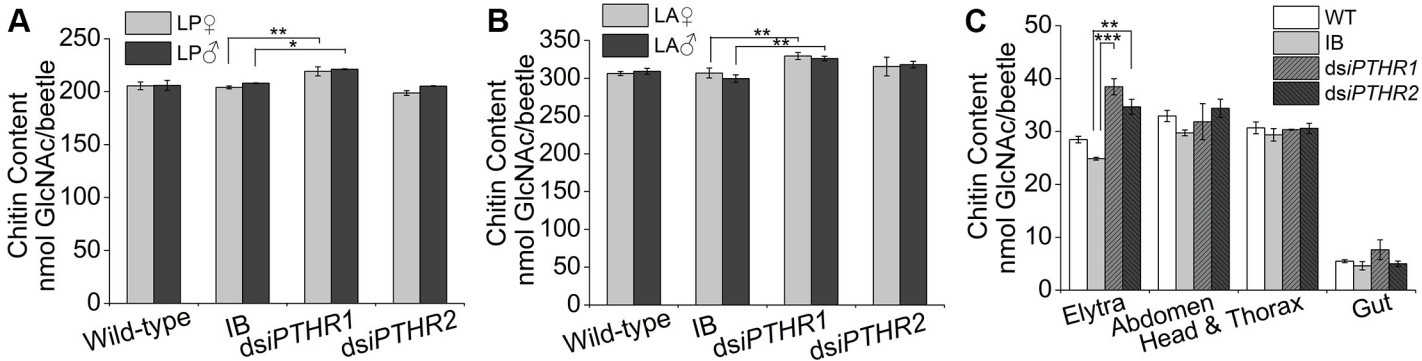

**Fig 9.** (A-B) Chitin content of the individuals in the LP (A) and LA (B) stage after dsRNA treatment. (C) Chitin content of the tissues of elytra, gut and epidermis in the adults after larval dsRNA treatment. The statistical results are for the treatments compared to the controls. *, p ≤ 0.05; **, p ≤ 0.01 and ***, p ≤ 0.001.

## Discussion

The initial purpose of this study was to identify the authentic ligand for the iPTHR, which is an ortholog of vertebrate PTHR. Similar phylogenetic distributions of the genes and the presence/absence of the neuropeptide candidates and iPTHR in the survey of different taxa allowed initial identification of iPTH. Activity of iPTH on the iPTHRs showed that iPTH does indeed acts a ligand for these receptors. RNAi of *iPTHRs* in *T. castaneum* suggests that iPTH signaling is required for cuticle maturation in the adult eclosion. It is interesting to find a function of iPTHR in exoskeletal development in *T. castaneum* while the major PTH system function in vertebrates includes skeleton development, although these findings could well be a consequence of analogous functional evolution.

The RNAi phenotypes for *Tc-iPTHR1* and *Tc-iPTHR2* together with the DE analyses after the RNAi provided insights into the functions of the iPTH system. The similarities of the RNAi phenotypes and the DE gene patterns after the RNAi of *Tc-iPTHR1* and *Tc-iPTHR2* indicate that although these two receptors may serve complementary functions, both are necessary as they are unable to compensate each other.

The mechanism behind the requirement of iPTHR for exoskeletal cuticle development is still enigmatic. Based on the DE data obtained 5 days after the injections of ds*iPTHR* (~prepupal stages), noticeable DE genes with biological significance for wing development are in two different categories: the genes for cuticle formation and signaling pathways. The major category of the DE genes consisted of genes involved in cuticle formation such as the proteins making up the structural components of the exoskeleton and chitin metabolism. Among the down-regulated genes, *chitinase 5* and *chitinase 10* were previously shown to affect degradation of old cuticle at the molting stage, ultimately leading to molting deficiency in *T. castaneum* [29, 30], while *knickkopf 3* and *tyrosine hydroxylases* are also well characterized for their essential roles in adult cuticle formations and sclerotization [31, 32].

Among the down-regulated cuticle proteins, it has been reported that some of the CPRs are highly expressed in the elytra and hindwings of newly molted *Tribolium* adults (4 and 9 respectively) [33]. Additionally, three CPAP (Cuticular Proteins Analogous to Peritrophins) proteins identified here have been reported to be expressed only in cuticle-forming tissues, and at relatively high levels in the elytra and hindwings of the 2 week-old *Tribolium* adults [34], in which *TC011140* and *TC009263* knockdown can result in adult lethality and adults with rough elytra unable to completely cover the abdomen, respectively [35]. A similar phenotype was also observed after RNAi of *Tc-iPTHR1* and *Tc-iPTHR2*. Among down-regulated extracellular

matrix proteins, keratin is a component of skin and epidermal appendages of vertebrates [36] and prismalin-14 [37] is a protein characterized as a novel matrix protein in pearl oyster.

In addition, up-regulated expressions of *Chitinase 8*, *9*, *16*, and *3-like* were found after RNAi of *Tc-iPTHRs*. Three of them (*Chitinase 8*, *9*, and *16*) belong to the Group IV chitinases, in which *Chitinase 8* contains only one chitin binding domain while *Chitinase 9* and 16 lack this domain [38]. It was reported there were no detectable abnormalities after RNAi of these three chitinases individually [29]. These results suggested that they are not essential while the phenotypes of over/mis-expressions of these genes are potentially disruptive in *T. castaneum* development. Likewise, a number of cuticular proteins were also up-regulated. These data together with the large number of down-regulated cuticular proteins indicate that the expressions of different sets of cuticular proteins were disrupted in the RNAi of *Tc-iPTHRs*. The conclusions drawn from the DE data analysis were also supported by cuticle contents changing in the RNAi pupae and adults, especially in the abnormal adults' elytra, as the direct causal factor of the phenotype during the eclosion.

The down-regulated genes involved in signaling pathways included the ecdysone metabolism genes *spook* (*Cyp307A1*) and *Cyp303A1* which has an unknown substrate. *Spook* is required in the early step of ecdysteroid synthesis for ketodiol formation in *D. melanogaster* and in *Schistocerca gregaria* [39, 40]. *Cyp303A1* in *D. melanogaster* is a gene required for bristle morphogenesis in cuticle metabolism [41] and its knockdown culminated in insects with a phenotype of defects in wing development at the adult eclosion [42]. Moreover, ecdysone-inducible genes [43, 44] *74EF*, *E78*, *DHR3*, *HR4*, and *E3* and *Ftz-F1* were also down-regulated (S1 Dataset). However, when the individuals with the *iPTHR* RNAi were examined in the stage where 20E is reaching its peak (3 days after pupation), we were unable to detect any differences in 20E immunoreactivity compared to the control. The current study and the knowledge available on ecdysteroid functions are insufficient to draw a conclusion as to whether iPTH signaling is required for ecdysteroid metabolism including the biosynthesis or inactivation steps.

In addition, other signaling molecule receptors were also down-regulated: *neuropeptide F receptor*, *rickets*, and *CG13579*, an amine receptor. Among these, *rickets* is well characterized as the receptor for bursicon that is required for cuticle sclerotization and wing expansion at the time of eclosion [45–49]. Involvement of neuropeptide F and biogenic amine in metamorphic development are as yet unknown.

The phenotype after RNAi of *Tc-iPTHRs* appeared at eclosion and early adult stage, while significant DE genes started to appear in the earlier prepupal stage. It is likely that the phenotype is a consequence of the culmination of accumulated physiological and developmental imbalances occurring in the earlier stages. In addition, although Q-PCR result showed a similar marked reduction of both *Tc-iPTH* and its receptors when the RNAi is applied, the inhibition may have not been sufficient for the phenotype in the RNAi of *Tc-iPTH* as the dsRNA of *Tc-iPTH* did not show any phenotype. We found that the RNAi for suppression of *Tc-iPTH* still displayed residual iPTH immunoreactivity, indicating that the *Tc-iPTH* RNAi did not provide an appropriate level of transcript knockdown and the leaking level of protein production was sufficient for the required levels of bioactivity. An alternative possibility is that there may be another ligand for PTH receptors that yet needs to be discovered in *T. castaneum*.

In summary, we identified a new insect peptidergic system, iPTH, which is the orthologous system to vertebrate PTH, showing that the PTH system is an ancestral signaling system dating back to the evolutionary time before the divergence of protostomes and deuterostomes, although it has been lost in Diptera and Lepidoptera. High expression at both mRNA and protein levels of the iPTH and its receptors in the gut and the CNS of *Tribolium* indicated that this neuropeptide may have important regulatory functions. We investigated this by RNAi and showed that iPTH system plays an essential role in cuticle maturation and fecundity in insects.

## Materials and methods

### Experimental insects and phylogenetic analysis

A laboratory colony of *T. castaneum* Georgia-1 line insects was used for all of these experiments and reared in whole wheat flour with 5% brewer yeast powder at 30˚C and 40% relative humidity on a 14h:10h light cycle under standard conditions [50].

Blast searches of iPTH and iPTHR sequences were made in the databases of NCBI: nr, RefSeq genome, genome, Expressed sequence tag, Whole-genome shotgun contigs, Transcriptome shotgun assembly, and of i5k workspace. The search algorithm and parameters were BLOSUM60 or PAM30 with the expected threshold of 50, followed by manual screens.

Transmembrane domains of the iPTHR sequences were determined by the on-line software TMHMM (v2.0) (http://www.cbs.dtu.dk/services/TMHMM/) [51]. Multiple sequence alignments of the amino acid sequences were performed using ClustalW [52]. The alignments were viewed and edited in BOXSHADE software (http://www.ch.embnet.org/software/BOX_form.html). The phylogenetic trees were constructed using MEGA 7.0 software using the Maximum likelihood method of the JTT model with 1000 bootstrapping tests [53].

### Q-PCR analyses for temporal and spatial expression patterns

For the stage-specific expression study, total RNAs were extracted from pools of multiple individuals in the following developmental stages: early embryo (EE, 1 day old, ~50 mg), late embryo (LE, 3 day old, ~50 mg), early larva (EL, 1 day old, ~50mg), late larva (LL, ~15 days old, 3 individuals), early pupa (EP, 1 day old, 3 individuals), late pupa (LP, 5 days old, 3 individuals), early adult (EA, 1 day old, 3 individuals), and late adult (LA, 10 days old, 3 individuals). For the tissue-specific expression study, total RNA of various tissues from approximately 100 LL and 100 LA was also extracted. The LL tissues included epidermis, gut, fat body, central nervous system (CNS), and in LA, besides the four tissues mentioned, accessory gland, testis and ovary were also collected. For the further expression investigations in the gut and epidermal system, separate foregut, midgut, hindgut, epidermis of head and thorax, epidermis of abdomen in the LL, LP, and LA stages were prepared, and in the LP and LA stage, the elytra were also collected. Reverse transcription was performed by using 1 μg total RNA. Q-PCR analyses were performed in MiniOpticon (Bio-Rad) with the SYBR green kit (TaKaRa, Japan). The relative mRNA levels were standardized to the control gene ribosomal protein S3 (rpS3, GenBank accession numbers CB335975), using the $2^{-\Delta\Delta CT}$ method [54]. The primers are listed in S2 Table.

### RNA interference

Systemic RNAi was performed as previously described [55]. Primers containing *Tc-iPTHR1* or *Tc-iPTHR2* sequences and the T7 polymerase promoter at the 5'-end of both the forward primer and reverse primer were used (S2 Table). The regions that were selected to amplify dsDNA showed low sequence similarity between *Tc-iPTHR1* and *Tc-iPTHR2* to minimize the potential cross activities of the RNAi. The PCR product was used as the template for dsRNA synthesis with the HiScribe T7 Quick High Yield RNA Synthesis Kit (New England Biolabs, USA). 30 beetles in two different stages, early 6th instar stage and 1th day pupal stage, were used for injections of 200 nl of 2 μg/μl dsRNA per beetle. Negative controls consisted of non-injection (wild- type group, WT) or injection of an equal volume of buffer only (IB group). RNAi efficiency was examined by Q-PCR using a pool of three individuals on the 5th day after dsRNA injections. The beetles were daily observed and mortality and phenotype were recorded. Egg numbers and hatching rate were obtained from 14–16 day old beetles of pupal RNAi. At least three replications were carried out for each RNAi phenotype count.

## GPCR assays

*Tc-iPTHR1* and *Tc-iPTHR2* cloning were based on TC008110 and TC010267 gene predictions (GenBank accession number XP_008192153.1 and XP_969953.1, respectively). The gene was cloned into pcDNA3.1+ and was used for heterologous expression in Chinese hamster ovarian (CHO) cells. Transient transfection was followed by an aquorine based functional assay as previously described [1, 26].

## Immunohistochemistry and microscopy

Tissue samples, CNS and gut from late larva, pupa(pharate adult) and early adult were dissected in PBS (pH 7.4). The tissues were fixed for 1~2 h at room temperature in 4% paraformaldehyde in PBS. After extensive washing, the tissues were incubated in primary antibody (1:1000 for anti-Tc-iPTH or 1:2000 for anti-Tc-iPTHR1) for 20 h at room temperature followed by incubation in secondary antibody (1:300 for goat anti-rabbit IgG-Alexa 488) overnight. Images were acquired with a Zeiss LSM 700. Anti-Tc-iPTH and anti-Tc-iPTHR1 was separately raised in a rabbit after three immunizations with the CTMPVGGGRIDPGRIamide (C-terminus 14 amino acids of iPTH) or SNEYIKWTRNNYATRQCamide (N-terminus 17 amino acids of iPTHR1) conjugated to keyhole limpet hymocyanin (KLH). The final bleed was purified by using the antigenic peptide. Pre-immune serum was used as the negative control.

The elytra phenotypes of *iPTHR1* and *iPTHR2* dsRNA injections at the EP stage were obtained for 10 day-old adults. The samples were dehydrated in an ethanol gradient of 60, 70, 80, 90, and 100 for 5 min each. Samples were coated with platinum and observed with a KYKY10008B electron microscope (JSM-5610, Tokyo, Japan) [56].

## RNA-seq analysis

Total RNA of 6 larvae was isolated at day 5 after dsRNA injections in the early 6th instar stage using a Trizol Reagent (TaKaRa, Japan) according to the manufacturer's protocol. The mRNA was enriched by using oligo(dT) magnetic beads. After fragmentation into ~200 bp lengths, it was used for the first strand cDNA synthesis by random hexamers. Library construction and sequencing was performed by BGI-Tech with Illumina HiSeq2000 [57]. Raw sequence data were submitted to the Short Read Archive (SRA) database of NCBI under the accession numbers SRR8559987 (ds*iPTHR1*), SRR8559988 (ds*iPTHR2*) and SRR1176913 (control).

The raw data were analyzed after filtering the low-quality sequences. Sequences were aligned to the *T. castaneum* genome (http://www.beetlebase.org/) using SOAPaligner/soap2. The expression level of genes from the RNA sequencing was normalized by the RPKM method (Reads Per Kilobase of exon model per Million mapped reads). The number of total mapped reads was approximately 7 million for each library. GO terms for each *T. castaneum* genes were obtained using Blast2GO (version 2.3.5) (http://www.blast2go.org/). Only those with more than 4× differences were kept for the DE analyses.

## 20E titer and chitin content assay

For measuring 20E titer, a single individual was homogenized in 500 μl high grade 100% methanol by using a plastic pestle, and then the samples were placed at 4°C overnight with gentle rocking. The tubes were spun at 13,000 rpm and the supernatant was collected. One more round of methanol extraction of the pellet was applied. Pooled methanol extractions were concentrated in the Eppendorf Centrifugal Vacuum Concentrator 5301 (Hamburg, Germany). Enzyme immunoassay for 20E quantification was performed as was previously described [28].

To detect the effect of the RNAi of *iPTHRs* on chitin synthesis, the insects treated with RNAi in the EP stage were collected on the 5[th] day after pupation and the 10[th] day after eclosion. Epidermal tissues (elytra, epidermis of abdomen, epidermis of head and thorax, and gut) of the 10[th] day adults were dissected. Chitin content of 3 pooled whole body or tissues of 10 individuals were assayed based on the method previously described [58]. Briefly, chitin is decomposed into chitosan under the action of high temperature (130°C) and alkali (14 M KOH), and then the chitosan could be depolymerized to glucosamine residues under a mixture of 10% $NaNO_2$ and 10% $KHSO_4$. After addition of $NH4SO_3NH_2$ and freshly prepared MBTH (3-methyl-2-benzothiazolone hydrazone hydrochloride hydrate), the mixtures were incubated at 100°C for 5 min in a water bath. Absorbance at 650 nm was determined in the Synergy H1 microplate reader (BioTek Instruments Inc., USA). Chitin content was expressed as a glucosamine equivalent according to a standard curve constructed by using known concentrations of glucosamine (Sigma-Aldrich).

## Supporting information

**S1 Fig. Maximum likelihood tree of the JTT model with 1000 bootstrapping of PTHRs in metazoan.**
(TIF)

**S2 Fig. Gene structure and deduced amino acid sequences of Tc-iPTHRs.** (A) Gene structure of the *Tc-iPTHRs*. The blank boxes represent the exons, and the line represents the introns. (B) Amino acid sequence alignment showing the similarities among PTHRs of invertebrates and human. Inverted letters with black and gray background are identical and similar amino acids in the sequence alignment with the 50% majority rule. Seven transmembrane regions have been marked above the sequence alignment. Star "*" indicates the conserved cysteine residues. (C-D) The sequence of the *Tc-iPTHR1* and *Tc-iPTHR2*, respectively. The underline is the 3' UTR sequences.
(TIF)

**S3 Fig. Expression patterns of *Tc-iPTHRs* in the gut and epidermal system.** These data supplement Fig 3 in the main text.
(TIF)

**S4 Fig. Negative controls of the immunohistochemistry of anti-Tc-iPTH antibody staining in the central nervous systems and gut in larval (A-D) and pharate adult (E-G) stage.** Preimmune serum was used as the negative controls of Tc-iPTH for immunohistochemistry and did not show any specific staining patterns. (H-O) Immunohistochemistry of Tc-iPTH in the late pupal stage with (H-K) or without (L-O) larval injection of dsRNA. Please note that compared to the IB pupae, the immunoreactivity in the ds*iPTH* pupae was significantly reduced, but still displayed remaining immunoreactivities for the Tc-iPTH.
(TIF)

**S5 Fig.** Negative controls of the immunohistochemistry of anti-Tc-iPTHR1 antibody staining in the central nervous systems and gut in larval (A, A', D), pupal (B, B', E) and adult (C, C', F) stage. Preimmune serum was used as the negative controls of Tc-iPTHR1 for immunohistochemistry and did not show any specific staining patterns.
(TIF)

**S6 Fig. Two-dimensional scatter plots showed differentially expressed genes in the control and ds*iPTHRs* treatments.**
(TIF)

**S7 Fig. Q-PCR confirmation of the expression profiles measured in the RNAseq experiment.** Twenty-two randomly selected differentially expressed genes between the control and ds*iPTHR1* or ds*iPTHR2* as determined by RNA-sequencing and Q-PCR.
(TIF)

**S8 Fig. Neighbor joining tree with 1000 bootstrapping of the down-regulated cuticle proteins after RNAseq.**
(TIF)

**S9 Fig. Immunoassay measuring 20E content in control, ds*iPTHR1* and ds*iPTHR2* pupae showed no significant differences in the immunoreactivity.** (A) Immunoreactivity for 3-day-old pupal stages for no injection, buffer injection, RNAi for *Tc-iPTH*, for *Tc-iPTHR1*, and for *Tc-iPTHR2*. (B) Immunoreactivity for 5-day-old pupal stages. Error bars are for standard errors of 3 biological replications.
(TIF)

**S1 Table. Fasta files showing PTH sequences captured in GeneBank database searches.**
(PDF)

**S2 Table. Primer sequences used in this study *Tc-iPTHRs*, *Tc-iPTH*, the internal control gene Tc-*rps3*, and other genes tested for the Q-PCR.**
(PDF)

**S1 Dataset. The genes down-regulated in RNAseq of the RNAi samples for *Tc-iPTHR1* and *Tc-iPTHR2*.**
(PDF)

**S2 Dataset. The genes up-regulated in RNAseq of the RNAi samples for *Tc-iPTHR1* and *Tc- iPTHR2*.**
(PDF)

## Author Contributions

**Conceptualization:** Jan A. Veenstra, Yoonseong Park, Bin Li.

**Data curation:** Jia Xie, Ming Sang, Donghun Kim.

**Formal analysis:** Jia Xie.

**Funding acquisition:** Jia Xie, Donghun Kim, Yoonseong Park, Bin Li.

**Investigation:** Xiaowen Song, Sisi Zhang.

**Supervision:** Jan A. Veenstra, Yoonseong Park, Bin Li.

**Validation:** Xiaowen Song, Sisi Zhang.

**Writing – original draft:** Jia Xie, Yoonseong Park.

**Writing – review & editing:** Yoonseong Park, Bin Li.

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
