## [Decision Letter · Decision Letter 0]

8 Jan 2020

Dear Dr Li,

Thank you very much for submitting your Research Article entitled 'A new neuropeptide insect parathyroid hormone iPTH in the red flour beetle Tribolium castaneum' to PLOS Genetics. Your manuscript was fully evaluated at the editorial level and by independent peer reviewers. The reviewers appreciated the attention to an important problem, but raised some substantial concerns about the current manuscript. Based on the reviews, we will not be able to accept this version of the manuscript, but we would be willing to review again a much-revised version. We cannot, of course, promise publication at that time.

If you decide to revise the manuscript for further consideration at PLOS Genetics, please aim to resubmit within the next 60 days, unless it will take extra time to address the concerns of the reviewers, in which case we would appreciate an expected resubmission date by email to plosgenetics@plos.org.

[LINK]

We are sorry that we cannot be more positive about your manuscript at this stage. Please do not hesitate to contact us if you have any concerns or questions.

Yours sincerely,

Liliane Schoofs

Associate Editor

PLOS Genetics

Gregory Barsh

Editor-in-Chief

PLOS Genetics

Reviewer's Responses to Questions

**Comments to the Authors:**

Reviewer #1: Comments:

Introduction

line [87] Add other references of relevant insect GPCR papers/reviews?

Results

[129] Change (Fig. 2A) to (Fig. 2A; Fig. S1). Figure S1 displays the gene duplication in insects more clearly.

[136] According to the supporting information file, Fig S1 only displays PTHRs and no iPTH sequences. Thus, there is either a reference to a Figure missing or a reference to literature.

[149-151] This is described in literature as characteristic of B-group GPCRs. Add this fact in your description/discussion, together with relevant references.

[166-167] Remove line (see the definition of early and late in the materials and methods), as it speaks for itself that the description would and should be in this section.

Discussion

[346-349] Agreed, this is a very interesting finding. However, how does this correlate with the neural projections of iPTH (Fig. 4) and the expression levels of both iPTH and both iPTHR (as described in Fig. 3 & 5)? Both receptors and iPTH itself seem to be mainly expressed in Gut and CNS (Fig 3.). There should be at least mention of this fact in your discussion, even if the effect would probably be indirect.

[354-355] You have not really proven that the receptors are unable to compensate each other, see comment line 460-462. Suggested experiment: RT-qPCR of both receptors in experimental tissues used of these experiments. As you already have the samples, this addition should be easily conducted. These results must be added in supplementary data. Or alternatively, show this fact explicitly in a figure separate from of your RNAseq data.

Materials and methods

[439] Define “multiple individuals” as this could describe any quantity, from 2 to infinity. The reader should be informed about the size of the biological pool by which one can determine the relevance of the described results.

[460-462] This potential has not been experimentally excluded, or it is not shown. One should. What kind of controls or additional experiments have been performed to confirm the data and to show that these are not due to off-target effects?

[466-467] This RNAi efficiency at RNA/protein level is not shown. Add to Supplementary data.

Reviewer #2: The review is uploaded as an attachment.

Reviewer #3: This paper describes the first identification of insect peptides that activate “orphan” receptors (iPTHRs) homologous to the vertebrate parathyroid hormone receptor (PTHR). On this basis, the authors designate these peptides “insect parathyroid hormones” (iPTHs). Interestingly, while vertebrate PTH regulates calcium homeostasis by modulating bone metabolism, iPTHs are implicated in regulation of exoskeleton maturation and fecundity. This is a major advance in assigning functional roles for iPTHRs and likely will further our understanding of ligand-receptor coevolution. The manuscript summarizes a great deal of work and is well presented. The bioinformatic analysis is excellent and RNAi knockdown creates phenotypes that involve integumentary deficits. Some questions remain should be addressed in the interests of clarifying the source(s) and targets of iPTHs.

Specific Comments

1. Designation of peptides as iPTHs. On one hand, it is understandable, since functions of iPTHs and vertebrate PTHs appear to be quite different. But why not simply refer to them according to species (e.g., TcPTH)? Perhaps the authors could explain the rationale for the designation of iPTH.

2. Sequences shown in Fig. E 1 fail to depict amidation at the C-terminus; it might be an idea to add a lower case “a” for amidation at the C-terminus of each sequence.

3. No mention of sequence similarity between iPTHs and vertebrate PTH. This reviewer found 47% similarity and 31% identity in a pairwise comparison with the consensus sequence shown in Fig. 1C. Addition of such information would complement analysis of sequence similarities of iPTHR and vertebrate PTHR.

4. Is there any evidence that iPTHs regulate hemolymph Ca levels, as is the case for vertebrate PTH?

5. Unlike the tissue source of vertebrate PTH, which is a well-defined gland behind the thyroid gland, iPTHs apparently originate from diverse sources (CNS, gut, etc). This is a matter of some uncertainty, however, since the antiserum used to locate iPTH-positive cells is not well characterized (see below). Hence, the possibility of cross-reactivity with other epitopes cannot be ruled out. Control experiments reported here are confined to negative results with pre-immune serum. This is not convincing. Additional controls could be: 1) determine if staining intensity decreases following RNAi of ligands and receptors. Although this has been shown to some extent in Fig. S4 for iPTH, convincing side-by-side comparisons with and without dsRNA are not provided. Another control could be pre-absorption of antiserum with ligand, following by staining, which should be negative. Likewise, no characterization of antisera raised against iPTHR is provided. Difficulties in specific immunohistochemical identification of GPCRs are well-known. Finally, it would be much more convincing if corroborating in situ hybridizations were included to demonstrate presence of the predicted transcripts for iPTHs and iPTHRs.

6. A second problem with the immunohistochemical data shown in Figs 4 and 5 is that the images are not described in enough detail. It is often difficult to discern which parts of the nervous system are depicted in Figs. They are simply referred to en masse; consequently, the reader cannot discern anything specific.

7. Any indication that iPTH is secreted by endocrine cells or glands, as is vertebrate PTH?

8. With regard to functionality of iPTH signaling, no mention is made in the abstract about reduction of fecundity following RNAi treatment.

9. Acronyms are not adequately defined in many instances; e.g. CPR family (line 325), DE analyses (line 512).

10. Abstract, line 39-40. Reference formatting is inconsistent and it is unusual to cite references in the abstract.

11. Lines 158-159. Fig 2B contains no information to support this statement of negative results with untransfected cells.

12. The manuscript requires further editing to correct English grammar and spelling.

**Have all data underlying the figures and results presented in the manuscript been provided?**

Reviewer #1: Yes

Reviewer #2: Yes

Reviewer #3: Yes

PLOS authors have the option to publish the peer review history of their article (what does this mean?). If published, this will include your full peer review and any attached files.

Reviewer #1: No

Reviewer #2: No

Reviewer #3: No

---

## [Decision Letter · Decision Letter 1]

20 Mar 2020

Dear Dr Li,

Thank you very much for submitting your Research Article entitled 'A new neuropeptide insect parathyroid hormone iPTH in the red flour beetle Tribolium castaneum' to PLOS Genetics. Your manuscript was fully evaluated at the editorial level and by independent peer reviewers. The reviewers appreciated the attention to an important topic but still identified some aspects of the manuscript that should be improved.

We therefore ask you to modify the manuscript according to the review recommendations before we can consider your manuscript for acceptance. Your revisions should address the specific points made by each reviewer.

[LINK]

Yours sincerely,

Liliane Schoofs

Associate Editor

PLOS Genetics

Gregory Barsh

Editor-in-Chief

PLOS Genetics

Reviewer's Responses to Questions

**Comments to the Authors:**

Reviewer #1: Several comments have been well addressed, but some are remaining:

Introduction

Line[97-98]: Described references are not added in the text, nor added in reference list. Please do so, check more carefully …

Results

Line[161] Add “,” between … binding pocket - and - as stabilization…

Discussion:

[441-445]: sentences are incoherent, revise wording.

More in depth: L442: Too vague: high expression of what? RNA, protein? Be more specific.

L443: Too vague: which additional functions? This does not add anything. Either describe which additional functions you imply, or don’t mention this.

[354-355]: Figure shows exactly what was asked for, but it should be shown to readers in order to make these claims. Add this figure either to Fig 7 or Fig 8 as an extra panel. Or if not shown, than the claim that the receptors are unable to compensate each other should be eliminated from your text.

Reviewer #2: I thank the authors for taking into account my remarks which I consider have been sufficiently addressed.

Some of the newly written sentences could be improved:

lines 58-59: Vertebrate parathyroid hormone (PTH) and its receptors have been extensively studied with RESPECT TO [the] THEIR function [of] IN bone remodeling and calcium metabolism.

lines 53-54: Moreover, RNAi of iPTHRs also led TO significant reductions in egg numbers and hatching rates after parental RNAi.

line 59-61: Insect parathyroid hormone receptors (iPTHRs) [were] HAVE BEEN previously described as counterparts of vertebrate PTHRs.

line 82-83: Functional studies of ancestral bilaterian neuropeptides have been successful by starting from the sequence similarities, e.g., vasopressin AND TRH [1-4]

line150: coincided with the [lack] ABSENCE of this specific

line 152: Maximum likelihood method with THE JTT model

line 220: neuronal cells and axonal projections in THE CNS

line 284: beetles arrested at the time of [ecolsion] ECLOSION

line 297: suppression of Tc-iPTH by the RNAi may [have not] NOT HAVE been sufficient to induce

lines 306-307: plots also showed THAT DE genes in the dsiPTHR1 and dsiPTHR2 treatments [to] have similar patterns

line 310: the correlation between dsiPTHR1 and dsiPTHR2 treatments [with a R-square showing 0.78] was obvious (R-SQUARED=0.78)

line 430: The CONCLUSIONS DRAWN FROM THE DE data ANALYSIS [together] were also supported by cuticle contents

line 460: [yet] STILL displayed [remaining] RESIDUAL iPTH immunoreactivity

line 472: THAT THIS NEUROPEPTIDE MAY HAVE additional functions.

line 805: "Two-dimensional scatter plots" instead of "Two dimensional scatter plots"

**Have all data underlying the figures and results presented in the manuscript been provided?**

Reviewer #1: No: See comments to authors

Reviewer #2: Yes

PLOS authors have the option to publish the peer review history of their article (what does this mean?). If published, this will include your full peer review and any attached files.

Reviewer #1: No

Reviewer #2: No

---

## [Editor Report · Decision Letter 2]

9 Apr 2020

Dear Dr Li,

We are pleased to inform you that your manuscript entitled "A new neuropeptide insect parathyroid hormone iPTH in the red flour beetle Tribolium castaneum" has been editorially accepted for publication in PLOS Genetics. Congratulations!

Yours sincerely,

Liliane Schoofs

Associate Editor

PLOS Genetics

Gregory Barsh

Editor-in-Chief

PLOS Genetics

Comments from the reviewers (if applicable):

**Data Deposition**

http://datadryad.org/submit?journalID=pgenetics&manu=PGENETICS-D-19-01925R2

**Press Queries**

---

## [Editor Report · Acceptance letter]

24 Apr 2020

PGENETICS-D-19-01925R2 

A new neuropeptide insect parathyroid hormone iPTH in the red flour beetle *Tribolium castaneum*

Dear Dr Li, 

We are pleased to inform you that your manuscript entitled "A new neuropeptide insect parathyroid hormone iPTH in the red flour beetle *Tribolium castaneum*" has been formally accepted for publication in PLOS Genetics! Your manuscript is now with our production department and you will be notified of the publication date in due course.

With kind regards,

Matt Lyles

PLOS Genetics

On behalf of:
